# Nest Box Bacterial Loads Are Affected by Cavity Use by Secondary Hole Nesters

**DOI:** 10.3390/ani13182989

**Published:** 2023-09-21

**Authors:** Agnieszka Zabłotni, Adam Kaliński, Michał Glądalski, Marcin Markowski, Joanna Skwarska, Jarosław Wawrzyniak, Jerzy Bańbura

**Affiliations:** 1Laboratory of General Microbiology, Department of Biology of Bacteria, Faculty of Biology and Environmental Protection, University of Łódź, Banacha 12/16, 90-237 Łódź, Poland; agnieszka.zablotni@biol.uni.lodz.pl; 2Department of Experimental Zoology and Evolutionary Biology, Faculty of Biology and Environmental Protection, University of Łódź, Banacha 12/16, 90-237 Łódź, Poland; michal.gladalski@biol.uni.lodz.pl (M.G.); marcin.markowski@biol.uni.lodz.pl (M.M.); joanna.skwarska@biol.uni.lodz.pl (J.S.); jaroslaw.wawrzyniak@biol.uni.lodz.pl (J.W.); jerzy.banbura@biol.uni.lodz.pl (J.B.)

**Keywords:** nest box use, bacterial load, parids

## Abstract

**Simple Summary:**

Nest box bacterial loads were studied during the prebreeding period in relation to nest box occupancy by Great Tits and Blue Tits in the previous season at two different study sites: an urban park site and a deciduous forest site. This analysis revealed that the interior bacterial load was higher in the nest boxes used in the previous season for breeding; however, the results were significant only in the forest area. In general, the bacterial load was significantly higher in the nest boxes placed in the forest study area than at the urban park site.

**Abstract:**

Among the environmental factors that affect bird nesting in nest boxes, the influence of microbial communities is relatively poorly understood. In this study, nest boxes used for breeding by secondary cavity nesters were sampled before the start of the breeding season to assess the bacterial loads of the nest box in relation to their previous year status. Different parts of the wooden nest box offer variable conditions for the development of bacteria. During the breeding season, the nest box entrance hole is wiped out by birds, delivering bacteria to their bodies, but during winter, it is exposed to unfavourable external conditions. The interior of the nest box, in turn, is also wiped by birds, but the conditions during winter are more stable there. Therefore, samples from the entrance hole and the interior of the nest box were taken at two different study sites: an urban parkland and a natural forest. We predicted that both the occupancy of the nest boxes during the previous breeding season by birds and the nesting sites would influence the bacterial load of the nest box. To verify this prediction, two categories of nest boxes were sampled at both study sites: nest boxes occupied by any of the two tit species (Great Tit or Blue Tit) in the previous season for breeding and nest boxes that had remained empty that year. The interior bacterial load of the nest box was higher in the nest boxes occupied in the previous breeding season, but only in the forest area. Furthermore, the bacterial load of both the entrance hole of the nest box and the interior was significantly higher in the forest study area in both occupied and unoccupied nest boxes. Our results show that the bacterial load of the nest box is positively related to the presence of nests in the previous breeding season and can vary between different sites.

## 1. Introduction

Microorganisms are ubiquitous in a variety of environments, and they inhabit all the available spaces there. One microhabitat, the avian nest, constitutes a unique environment for a variety of them [1,2,3,4]. Bacteria that colonise avian nests and spaces which birds can potentially choose to breed in affect their avian hosts in positive and negative ways [3,5,6]. Some of the bacteria that colonize avian nests are commensals that feed on nest components consisting mainly of organic matter, but many bacterial species, including pathogenic strains, can exert a negative impact on birds (see [6] for a review). Recent research has shown that the nest microbiota plays an important role in mediating the life histories of birds. In particular, microorganisms shape the microbiome of their hosts and play a key role in food digestion (see [7] for a review) and pathogen defence [8,9], as well as influencing the quality of the plumage [10,11]. There is also a growing body of evidence showing the importance of microbial species for traits of the avian condition and, consequently, for birds’ reproductive success [2,3,4]. However, the relationships between the nest bacterial load and life history traits of birds are relatively poorly understood, mainly because the majority of studies on animal-associated microbiota have been conducted on captive animals [12]. Little is still known about the complex relationships between different species of birds and their microbiota under particular environmental conditions in the wild. Given the ubiquity of bacterial communities and the complexity of host–microbiota interactions, it is important to document the differences in both the abundance and diversity of microorganisms in different habitats [13,14].

Among the wide variety of locations where different species of birds breed, nests located in cavities are unique, as cavities constitute a particularly suitable environment for different types of microorganism [1,3,15,16]. The special characteristics of nest boxes (and other cavities, including natural ones) result from the maintenance of relatively stable physical conditions, such as humidity and temperature, inside the boxes, which make them suitable for microbial colonization and growth [2,17]. In addition, nest boxes are frequently used for breeding in several consecutive seasons, ensuring the steady delivery of organic matter (feathers, peeled fragments of the epidermis, or excreted faeces) essential for microbial growth. It should also be mentioned that outside the breeding season, several species of birds use nest boxes as roosting places during the autumn–winter period [18]. Furthermore, the available data suggest that the prevalent practise of cleaning nest boxes after the breeding season does not remove all the litter, leaving enough residues for bacterial growth [4].

In general, it is highly probable that the bacterial load of the nest box itself depends on a variety of environmental characteristics of a particular location. The physical properties of habitats include several characteristics such as humidity, thermal conditions, the chemical properties of the soil, and probably many other characteristics that influence the assemblages of bacteria on different spatial scales (see [13] for a review). Furthermore, the richness of plant species in a particular habitat can also play an important role [19]. All of this is of special importance when anthropogenic effects in altered habitats are involved. For example, urbanisation is known to have an impact on nest material selection, which can affect nest sanitation, thus altering the nest environment available for microorganisms [20,21,22]. In addition, urban metal pollution has been found in nest material, and it can negatively affect the reproductive output of birds (e.g., [23]). Therefore, since studies on natural microbial diversity in specific places such as nest boxes placed in different habitats are still rather scant [24], there is a need to focus on microorganisms colonizing nest boxes which are used not only for breeding but also as the roosting sites of many avian species.

For the reasons mentioned above and given that, in our previous study, we demonstrated a negative influence of the bacterial load on the physiological condition of wild birds [4], we conducted a study on two sets of wooden nest boxes that are used by two species of secondary cavity-nesting birds, the Great Tit (*Parus major*) and the Blue Tit (*Cyanistes caeruleus*), at two different study sites: an urban parkland and a deciduous forest. For several years in our study area, both species have interchangeably used nest boxes for breeding in consecutive breeding seasons; however, in a particular year, some of the nest boxes may remain unoccupied throughout the breeding season. This may potentially lead to variation in the abundance of bacteria in nest boxes that probably remains on particular level all year round, including at the end of winter, directly before the onset of the breeding period. The second potential source of variation in the bacterial loads in the nest boxes may be due to habitat differences between the two study sites. It is well-known that urban parks are simplified and highly transformed versions of natural habitats for numerous bird species, including secondary hole nesters such as parids. The different physical conditions that prevail at either site (temperature, humidity, soil characteristics, and similar characteristics) can probably shape the microbial communities in nest boxes. Moreover, it is likely that not only the specific conditions themselves in these two sites matter. In the case of the rich, seminatural forest site, the conditions are probably more stable over the years. On the other hand, the urban parkland area offers rather unstable conditions due to human-made interventions. This creates quite different conditions for the bacterial assemblages that inhabit both sites.

Nest boxes themselves are structures that create many microenvironments for bacteria. The various surfaces of the nest box differ with respect to the intensity of contact with birds that carry microorganisms on their feathers, beaks, or legs. For that reason, particular parts of nest boxes such as the entrance hole are wiped out by birds regularly during the breeding season (by parents and also by nestlings at the end of the rearing period) and by individuals spending nights there in the winter. The interior of the nest box, in turn, is in contact with adult birds during the period of nest construction and subsequently with nestlings that have direct contact with the inner walls. Particular parts of nest boxes are also exposed to external conditions such as temperature, humidity, or insolation to different extents, which creates more or less stable microhabitats for microorganisms, including bacteria. It is very likely that during the winter months, the entrance hole area, which is directly exposed to weather conditions, may offer less favourable conditions for bacteria than the nest box interior. Therefore, we decided to sample both the interior and the entrance hole of the nest box. We estimated the interior bacterial loads of the nest box and the entrance hole to verify the following prediction: since the two parid species are similar in terms of nest type material and other breeding characteristics, the nest boxes used for breeding by any of the two parid species during the previous year (later, ‘occupied in previous season’) would have higher bacterial loads than the nest boxes that had remained empty that year (later, ‘unoccupied in previous season’). The second objective of this study was to test whether there was a significant difference in the bacterial load between the two study sites: the urban parkland site and the forest site.

## 2. Materials and Methods

### 2.1. Study Sites

This study was carried out in 2020 in two different habitats: an urban parkland and a deciduous forest. The study sites are located ~ 10 km apart and are separated by the city of Łódź. The urban parkland study site (51°45′ N; 19°24′ E) consists of Łódź Botanical Garden, which covers a total area of approximately 67 ha. It is primarily of anthropogenic origin and has fragmented tree and bush cover with very few remnants of natural stands, including birches (*Betula pendula*), beeches (*Fagus sylvatica*), and numerous alien species planted intentionally by garden administrators [25]. The forest study site (51°50′ N;19°29′ E) is an area of approximately 145 ha located in the interior of a rich, mature, mixed deciduous forest called the Łagiewniki Forest (1250 ha in total). Oaks (*Quercus robur* and *Q. petrea*) are the dominant tree species in the forest. Both study sites were supplied with standard wooden nest boxes, each made of pinewood and with the same diameter of the entrance hole, 29 mm (200 in the parkland area and 300 in the forest area). After each breeding season in mid-October, each nest box at each the study site was cleaned with a wire brush, so that there were no visible nest remnants.

### 2.2. Bacterial Sampling

In the first half of March 2020, 40 nest boxes (20 at each study site) were randomly chosen for bacterial sampling. Neighbouring nest boxes were excluded from the study. Also, given that the nest boxes at both sites are at least 50 m apart from each other and only a proportion of them are occupied for breeding each year (up to a half in the parkland and a 25% in the forest), the nest box samples were independent of each other. Approximately half of the nest boxes at each study site had been occupied by one of the two tit species (Great Tit or Blue Tit) in the previous year for breeding, while the other half had remained empty that year (5 occupied by Blue Tits and 6 by Great Tits in the parkland and 4 occupied by Blue Tits and 6 by Great tits in the forest). Since the nest boxes are used interchangeably between years by both species, the data for Great Tits and Blue tits were pooled. Using disinfected latex gloves, an entrance hole was swabbed in each nest box with circular movements for 30 s with a sterile cotton swab previously moistened with sterile phosphate-buffered saline (PBS, pH 7.2; Adlab, Poland). Subsequently, a restricted area of ca. 2.5 cm^2^ of an internal rear wall of each nest box, directly next to the entrance hole, was swabbed in the same manner. One sample per entrance hole and another sample per inner wall of each nest box were taken (in a standardized manner with respect to time and area). In total, 80 samples were taken from the field. Some of them were inappropriate for further analysis due to technical problems, and finally, 71 were analysed. Only nest boxes with no or very little faeces at the bottom were sampled.

### 2.3. Lab Procedures

Once in the laboratory, bacteria from the swabs were transferred to a solution in the same manner as in our previous study on the bacterial load (see [4] for technical details). The bacteria were cultivated and incubated on Tryptic Soy Agar medium (TSA) for 48 h at 37 ± 1 °C and then for an additional 48 h at 25 ± 1 °C, and the colony forming units (CFU) (Appendix A) were counted. The results were expressed as CFU/mL.

### 2.4. Statistical Analyses

Because the bacterial loads of the nest box entrance hole and the nest box interior variables were expressed as CFU/mL, the values ranged from hundreds to tens of millions. Therefore, the raw numbers were normalized through ln transformation prior to analyses. Homoscedasticity was checked using Cochran’s test, and then two-way ANOVA was used to test whether the study area and the occupancy of the nest box affected the nest box entrance hole bacterial load in the study year. The interaction term was also calculated. This approach was repeated for the nest box interior bacterial load. Interactions, both significant and non-significant, were included in the models. The analyses were performed using Statistica ver. 12 software [26].

## 3. Results

The mean entrance hole bacterial load differed significantly between the study areas and was higher in the forest study area for all the nest boxes (occupied and unoccupied) (Table 1 and Table 2, Figure 1). The interior bacterial load was significantly higher in the nest boxes occupied in the previous year, but this difference was significant only in the forest study area (Table 1 and Table 2, Figure 2).

## 4. Discussion

We found that the interior bacterial load of the nest box was higher in the nest boxes used for breeding in the previous season, but this difference was significant only at the forest site. We also found that both the bacterial load of the entrance hole and the bacterial load of the nest box were significantly higher in the forest than in the parkland, in the nest boxes both occupied and unoccupied in the previous season.

The occupancy of the nest box in the previous year affected the bacterial load of the nest box. Our results are ambiguous on this point, since this effect was significant in the case of the interior of the nest box but not in the case of the nest box entrance and only in the forest study site. It is not clear why this effect was only found at one study site. A possible explanation may be related to the intensity of use of the nest boxes as roosting sites during the autumn–winter period. It is well-known that wintering tits spend the night in nest boxes ([18], own observations) and thus probably transfer bacteria there. In the parkland site, in contrast to the forest area, both species are more abundant during winter (own observations) and probably use nest boxes as roosting sites more intensely. If this assumption is true, it should at least partially eliminate the expected difference in the bacterial load between the nest boxes that were occupied and unoccupied for breeding. However, it seems that this effect is not strong enough to nullify the striking difference between the study sites. Yet, for this study, we chose nest boxes with no or very little traces of faeces in the bottom. Despite this, we still do not have any quantitative data on the intensity of use of nest boxes as roosting sites out of the breeding season and, therefore, we cannot make any plausible conclusion on this issue. There is also the possibility that in March, some individuals start to inspect nest boxes before nest building starts; however, we expect that these short visits are evenly distributed between boxes and do not impact our results substantially.

We revealed a clear difference in the interior bacterial load of the nest boxes between the deciduous forest and urban parkland. This result suggests that the bacterial loads in nest boxes may be strongly habitat-dependent. The interior of the nest box is probably readily colonized by a variety of microorganisms shortly after the placement of the nest box in a particular environment. Birds play the main role of transferring microorganisms to nest boxes when visiting them, as well as to other various cavities as places potentially suitable for breeding or roosting [19,24]. Bacteria are found in the beaks and toes; however, the communities of the plumage are the most important in this context, since bacteria are both most numerous and most diverse in feathers [24,27]. In our study system, the nest boxes were used mainly by the Blue Tit and the Great Tit. Both species forage mainly on leaves and twigs. Plants, as a source of bacteria, are suggested to play an important role in the transmission of these organisms to the plumage [27]. This impact of the plant host on the bacterial microbiota was shown between predators and prey in the trophic networks of Blue Tits [28]. Given that our study sites differ markedly from a floristic perspective, with deciduous forests being richer in plant species than urban parkland, it seems very likely that the forest site may maintain richer bacterial communities [19]. Researchers showed such a pattern of habitat-related differences in bacterial density and species richness. They revealed that although the number of phylotypes per bird was higher in a coniferous habitat, the bacterial densities were higher in a deciduous habitat. This finding is also supported by the results presented in [29], where the authors suggested that microbial activity was positively correlated with plant productivity. Similarly, the authors of [27] found that microorganisms from particular groups were more abundant in plumage sampled from the American redstart (*Setophaga ruticilla*) in wet versus dry habitats. This is in line with our results, since rich, mature forests retain more moisture than urban parklands, with their many open areas, no or little tree cover, and, therefore, openness to direct sunlight. However, the richness of the plant species is not the only difference between the two sites. The Botanical Garden is a place where various agrotechnological works are conducted throughout the year (own observations). These procedures include intensive grass mowing, branch cutting, and the application of different chemicals, such as insecticides, which probably affect different organisms, including bacteria that are present not only in plants but also in soil. For example, in 2013 and 2014, a large-scale molluscicide treatment was applied in the garden to eradicate the invasive Spanish slug *Arion vulgaris* (see [30] for details). Presumably, the regular use of chemicals in parkland, contrary to a forest area, can disturb microbial communities, including the sheer number of bacteria, in different ways. Furthermore, other factors, such as the presence of bird aggregations, influence soil properties and the microbial community in the soil [31], which may be important in the garden area where, during the autumn–winter period, wintering birds gather in relatively large numbers (own observations). For the above reasons, the microbial profile of the soil itself may play an important role in the acquisition of plumage bacteria [27,32,33]. This may be important in the context of this study, since the Great Tit forages on the ground relatively frequently ([34], own observations), which means that the birds acquire bacteria directly from the soil and then carry them into nest boxes. Therefore, in general, we expected that the bacterial load in the forest nest boxes would be more similar between years because of the greater stability and complexity of this site. On the other hand, the bacterial load in the urban park nest boxes would randomly vary across years because some bacteria cannot survive under such variable conditions or would survive at a lower level than in the forest. Since we analysed one year only, we cannot prove such an interpretation. Therefore, long-term studies on this issue are needed.

In addition to basic habitat characteristics, including plant species composition and soil characteristics, there are probably other factors that can contribute to bacterial loads in nest boxes. It is known that different taxa of animals visit nest boxes for a variety of purposes. Many invertebrates (snails, spiders, or insects) are opportunistic species that use nest boxes year-round as shelters or roosting places or to build nests and raise their offspring (i.e., wasps) ([35,36], own observations). Invertebrate species colonizing nest boxes carry their bacteria acquired from the environment. Since the forest area is a more diverse habitat than the parkland area in terms of plant species composition, it is likely that the invertebrate assemblies are also richer in the forest. The next potential factor is the presence of particular mammal species, which visit nest boxes for two main reasons. In both our study areas, brown long-eared bats (*Plecotus auritus*) and noctules (*Nyctalus noctula*) occasionally use empty nest boxes to breed and roost during the summer/autumn period, and they are more frequent in the forest (own observations). The pine marten (*Martes martes*)*,* in turn, is a predatory species that tries to reach the nest in the nest box and grab an adult bird or nestlings with its paws as prey. In some breeding seasons, the predation rate was very high in the forest [37] but not in the parkland, where marten predation occurred only in exceptional circumstances (own observations). Both bats and martens carry bacterial flora, which is transferred to nest boxes; however, in the case of martens, this transfer is mainly restricted to the entrance hole.

## 5. Conclusions

Our study demonstrates that the observed difference in the bacterial loads of the nest boxes may have been affected by their status as occupied by breeding birds in the previous season, which may have resulted from habitat-related differences between the two study sites. The most important factors are probably associated with the richness of plant species and the physical properties of the soil, with additional factors resulting from the specificity of the two study sites. However, taking into account the complex and yet relatively weakly understood relations between microorganisms, their hosts, and the environment, other factors probably play a role, and we are still far from understanding this issue. It is also clear that other site-specific characteristics, such as the presence of ectoparasites, various assemblages of invertebrates using nest boxes as opportunist species, and other external factors related to the structure of a particular site, may influence the environment for bacteria. We suggest that in future studies, these parameters should be taken into account and used in multi-factor modelling. Furthermore, we should interpret our results with care, since we do not have data on the richness of bacterial species at both study sites. We should stress here that our study, conducted in a single year and in two types of environment with only a relatively low number of nest boxes, does not allow for generalisations, and the results should be considered as tentative. However, we think that these results are valuable, since our study suggests that the bacterial load may be a potentially important factor for the life history traits of secondary cavity-nesting birds, because it may vary considerably between habitats. Furthermore, we showed that at the end of winter, directly before the breeding period, the bacterial loads in nest boxes are at relatively high levels and remain one of the significant factors that affect the nesting environment for secondary cavity-nesting species. Therefore, since studies on cavity-nesting birds with the use of artificial wooden nest boxes are widely conducted across a broad geographical range and a variety of habitats and given that the bacterial load can affect the host’s physiological health state, we consider our study prospective, as a promising foundation for future research.

## Figures and Tables

**Figure 1 animals-13-02989-f001:**
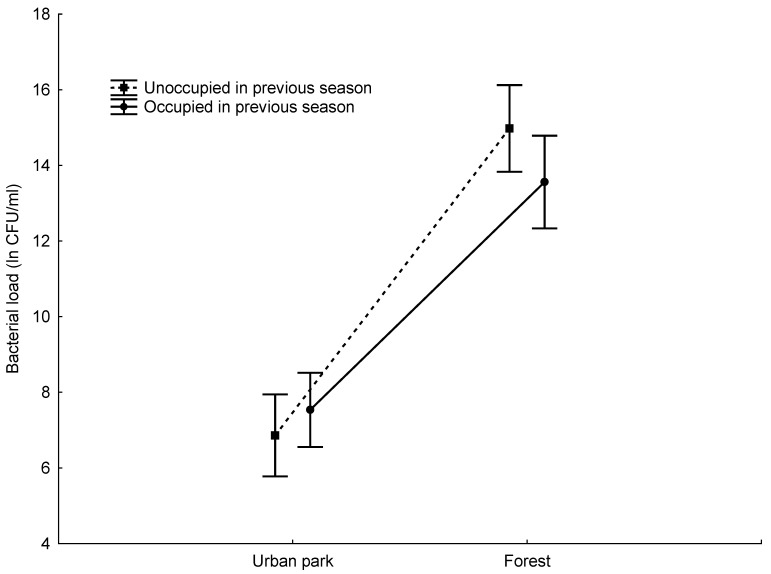
Mean (±95% confidence interval) entrance hole bacterial load in the nest boxes occupied and unoccupied in the previous season in the two study areas.

**Figure 2 animals-13-02989-f002:**
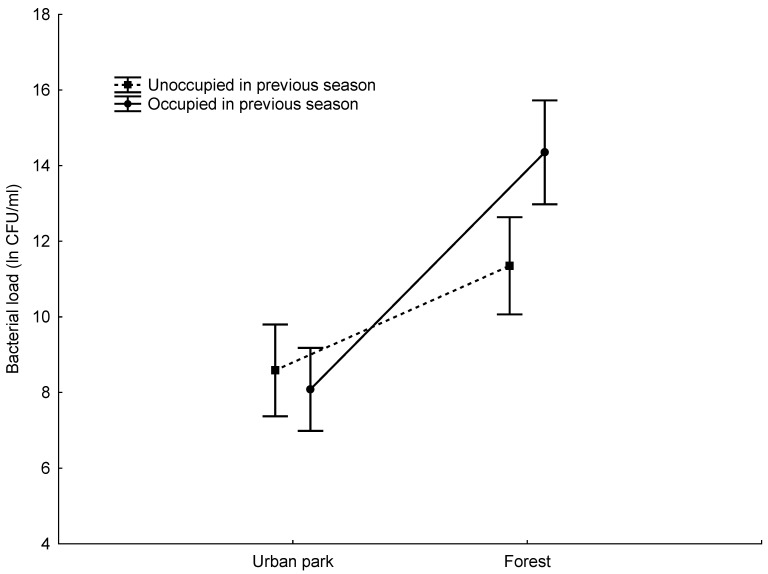
Mean (±95% confidence interval) nest box interior bacterial load in the nest boxes occupied and unoccupied in the previous season in the two study areas.

**Table 1 animals-13-02989-t001:** The mean, minimum, and maximum values of the entrance hole bacterial load and nest box interior bacterial load (CFU/mL) in the parkland and forest study areas in both nest categories (occupied and unoccupied in the previous year). Values are ln-transformed. SD values are given in parentheses.

	Parkland			Forest		
Entrance Hole Bacterial Load						
	Mean	Min	Max	Mean	Min	Max
Occupied	7.54 (±2.01)	5.30	10.37	13.56 (±1.91)	10.40	15.40
Unoccupied	6.86 (±0.65)	5.99	7.78	14.77 (±2.04)	12.82	17.97
Nest box interior bacterial load						
Occupied 8.08 (±2.01) 5.30 12.47 Unoccupied 8.58 (±1.16) 6.68 10.82				14.35 (±1.85) 10.82 16.07 11.35 (±1.98) 8.99 15.40		

**Table 2 animals-13-02989-t002:** Summary of the bacterial load in the two-way ANOVA of the entrance hole bacterial load (top) and the bacterial load in the nest box (bottom). The effects of the study area, nest box occupancy in the previous season, and the interaction between these factors are given. Interactions are marked with a * symbol. Cochran’s test for heteroscedasticity: entrance hole (C_3_ = 0.40, *p* > 0.05) and nest box interior (C_3_ = 0.31, *p* > 0.05).

Factor	Df	F	*p*
Entrance hole bacterial load			
Intercept	1; 31	1550.001	<0.001
Study area	1; 31	168.089	<0.001
Previous season occupancy	1; 31	0.465	0.500
Study area * previous season occupancy	1; 30	3.865	0.064
Nest box interior bacterial load			
Intercept	1; 31	1199.654	<0.001
Study area	1; 31	54.575	<0.001
Previous season occupancy	1; 31	4.165	0.049
Study area * previous season occupancy	1; 30	8.207	0.007

## Data Availability

The dataset generated and analysed during the current study is available in the Figshare repository, 10.6084/m9.figshare.23986410.

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
