# Peer review of "Nest Box Bacterial Loads Are Affected by Cavity Use by Secondary Hole Nesters"

_animals, 2023, doi:10.3390/ani13182989_

Round 1

Reviewer 1 Report

This work makes a very valuable contribution to the understanding of bacterial load in artificial nests.

It would be interesting for future studies to check what is the influence of bacterial loads for birds condition and their reproductive success. In particular, to check whether the value of bacterial load in prebreeding season has the influence on birds. Nevertheless, such analysis should takes into account also other factors, like specific habitat parameters and ectoparasites (multi-factor modeling).

Other interesting topic is to compare nestboxes and natural cavities in relation to bacterial load. It is especially important for birds protection which is often based on hanging of nestboxes.

Line 117-118 – Am I right that nest-boxes in both study sites were cleaned in October?

Figure 1 and 2 - Use the same hatching for unoccupied and occupied nest-boxes.

Table 1 - Correct bolds.

Discussion - The discussion is comprehensive and takes into account many research results that may be responsible for the results obtained in the assessed work. However, the work itself does not check the hypotheses discussed. It only gives pure results.

Line 178-179, 217-219 It would be highly appreciated to deepen the analysis by collecting data on habitat parameters and other factors which can influence the bacterial load and to use it in multi-factor modelling.

Line 227 – Mistakes in species names. Should be: Plecotus auritus, Nyctalus noctula.

Author Response

  1. We agree with that point. See, for example, our previous study (ZabÅ‚otni et al. 2020), where we showed that bacterial load can negatively affect nestlings’ physiology. In this study we showed differences in bacterial loads between nest boxes used and not used for breeding in the previous year, but also the difference between an urban park and the forest. We did not study the possible impact of bacterial load in March on reproductive success in subsequent spring, but possibly the relationship exists. We highlight this in conclusions. Undoubtedly, a wide array of environmental factors shapes the bacterial load (including ectoparasites, structure of soil, floral species diversity, and others). Since we do not have specific, solid data on these issues we could not insert them into multi-factor analyses, however, we suggested in conclusions that the above mentioned factors should be taken into account in future studies.
  2. That is true, however, our study system is based on nest boxes only. In addition, we do not know of any studies on the bacterial load in natural cavities occupied by birds. Given the variability of types of natural cavities, the study on bacteria that inhabit them would be difficult to make and results probably ambiguous. Nevertheless, we agree that such a comparison would be valuable.
  3. That is correct, we cleaned all the nest boxes in both study sites in October. We clarified that issue in Methods.
  4. We unified the confidential interval bars as solid lines for both the unoccupied and occupied nest boxes.
  5. The bolds are corrected.
  6. We broaden the discussion in the point of complexity of bacterial load in the forest versus urban park.
  7. As we noticed earlier, we cannot insert into analysis data on habitat parameters, since our data were collected in a particular time (March 2020). We suggested such an analysis in future studies in conclusions.
  8. Mistakes in species names are corrected.

Reviewer 2 Report

Dear authors,

The manuscript is well written, and major studies were cited. Results and methods are well presented.

However, I´m concerned with two major problems:

First, the authors conducted the study in only two sites: a parkland and a forest fragment. This small (the smallest possible) number of study sites weakens the major conclusion - that the bacterial load varies with habitat. You should have replicates (4-5 forests and 4-5 parklands) to support this conclusion.

Second, the number of studied nest boxes per habitat appears to be too small. If you have 500 of them installed in the study sites, could you increase your sampling ?

Because of these two major concerns, my suggestions is rejection. 

I´m sorry if I disappoint you, but I wish success when trying a new submission in this or another journal, as the topic of your study is interesting and important.

It is nearly ok.

Author Response

  1. We do not agree with this objection. We modified the title of our study and the sequence of objectives of the study at the end of introduction to emphasise that nest box occupation by birds in the previous season was of our main concern and the two habitats were rather replications of our main objective. We found, of course, between-habitat difference in bacterial load, and we appropriately discussed it (see also replies to Reviewer#3). Our results show a difference between a particular urban park and a particular forest, and we do not state that other parks or other forest types have similar characteristics. However, we still think that our study is valuable and shows an additional, relatively rarely studied factor that can influence the fitness of birds.

2.  We only partially agree with this. The sample sizes were small but adequate to conduct proper analyses and obtain significant results. We also insert in conclusions some notes on additional environmental characteristics.

Reviewer 3 Report

Review of the paper „Nest box bacterial loads are affected by habitat and cavity use by secondary hole-nesters”

The proposed manuscript compares the bacterial load in the nest box entrances and interiors between forest and urban park, taking into account whether great tit or blue tit nested in a given nest box in the previous breeding season. The authors found that bacterial load only in the nest box interior affected the bacterial load in the following breeding season, which concerned only a forest. Although the result seems interesting, in my opinion, the manuscript lacks a good ecological hypothesis that takes into account the complexity of the forest and urban park. I would like to point out that the bacterial load was significantly higher in the forest than in the urban park, which can be explained by the greater complexity and diversity of the forest. If so, it is difficult to expect that the diverse and complex bacterial load will disappear after one year as easily as the simplified version in urban parks (see below). A serious complaint is that it is difficult to understand and explain the comparison between the bacterial load of the nest box entrance and the back wall from the inside – the bacterial load has a different origin in these two places and different exposure to contact with birds and external environmental factors. The description of statistical analysis is too short.

Simple summary – in L15 something is missing

Abstract

L20-21 – the reason for the division of bacterial load into the entrance hole and nest box interior should be explained already in the Abstract

As for the ecological justification why only the bacterial load inside nest boxes occupied in the previous year affected the bacterial load in the current year, the most reasonable seems to be the influence of external factors, which certainly contributed to the fact that bacteria did not survive on the outer surface of nest boxes, however, in my opinion, these are not just weather factors mentioned by the authors of in the introduction. It seems to me that forest always provides more stable and more complex conditions than urban parks. Therefore, I would expect that bacteria in forest nest boxes should be more similar between years and more complex, and hence the impact on bacterial load in the current year in forests as opposed to urban parks. In urban parks, I would expect more random bacterial loads in different years and simplified conditions in which some bacteria are unable to survive. I did not find such an interpretation in the proposed manuscript.

Introduction

There is a lack of knowledge background comparing the stability and complexity of external environmental conditions in forests and urban parks (see above).

Materials and methods

I don't understand why the middle part of the rear wall was used to download the bacterial load reflecting the nest box interior. I would rather expect to collect samples from a place comparable to the hole entrance in terms of the frequency of touching by birds, i.e. the inner wall of the bottom of the nest box. The bacterial load on the rear wall in the middle part is incomparably less wiped by birds than at the entrance. Please note that the entrance hole is mostly worn down by adult birds, and the roaring wall is not even about the chicks, but about the nesting material touching that wall. The methods lack an ecological explanation for the selection of sampling sites in nest boxes.

Statistical analysis is poorly written, e.g., there is no information concerning the test for the homogeneity of variance that should be prepared before the analysis of variance.

L139 – please, explain the ln transformation

There is no information on data heteroscedasticity.

As I understand it, the sample size was small – how did the authors solve the problem of model overfitting?

Results

L152-153 – what was the direction of this relationship? It should be written here.

Discussion

Please, discuss the complexity of bacterial load and habitats in a forest compared to a simplified urban park.

Supplement – I do not understand whether the database is without any explanation (i.e., legend) and in the paper there is no feedback on the missing data in the two sites and the way they were treated in statistical analysis. I would suggest a table of results in a text document and descriptive statistics rather than raw data.

Please, check the English language carefully, especially the wording and grammar. Use past tense when you describe your results.

Author Response

  1. The simple summary was corrected.
  2. We insert such an explanation, and because we agree with your remarks on that point, we therefore add a more detailed description with proper justification of the choice of the two areas of the nest box for sampling at the end of introduction (detailed reply, see point 4). However, we only partially agree that bacteria survive worse in that inner circle of the entrance hole than on the back, inner wall, in fact our results show that the values are similar or even greater than inside the box, at least in unoccupied nest boxes in the forest. We agree that not only weather alone shapes bacterial loads in nest boxes and we add a more detailed explanation in the introduction. Your suggestion that bacteria in forest nest boxes should be more similar between years and more complex in contrary to urban park is valuable, and we add it in the discussion with objection that since we analyse one season only, this should be treated with caution.
  3. Introduction: we completed it as you suggested (see above).
  4. Materials and methods: We do not agree with that. As a middle part of rear inner wall, we understand the place exactly opposite to the entrance hole. In fact, it was imprecisely described in the methods, and we corrected this. It is in fact important, since the upper edge of the nest material does not reach this point and simply must be lower than the entrance. For that reason, this area of the inner wall is appropriate for sampling and is wiped out by birds (both adults and chicks) during the nestling rearing period. The bottom of the nest box has contact with birds only during the beginning of the nest building stage. Therefore, we think that these two parts are appropriate for bacterial sampling: inner wall has contact with adults and chicks, entrance hole mainly with adults but at the end of nestling period also with older nestlings.
  5. We agree with that and used Cochran’s test for the homogeneity of variance, which is described in the Methods section.
  6. Transformation was explained.
  7. Since analysis of homogeneity of variance was used (both for the entrance hole and inner wall), the results were placed in the caption of Table 2.
  8. It is corrected.
  9. The discussion and conclusions were supplemented as you suggested.

Supplement: we add an explanation to our data set. Since descriptive statistics are in the text in Table 1, we did not add them here.

Reviewer 4 Report

In the introduction, I would expect all the potential factors affecting the amount of bacteria, so the diet and foraging way are probably important.

Probably authors do not want to show all the results (comparison between great and blue tit) especially in the context of different foraging method, but nevertheless, the results are important and should be published.

Author Response

  1. In the introductory exploration (not included in the text) we checked the potential influence of the species (great tit versus blue tit) inhabiting the nest boxes, but the results were nonsignificant for both the entrance hole and the nest interior. Therefore, we did not discuss the possibility of different foraging strategies.

Round 2

Reviewer 1 Report

The authors' explanations are sufficient.

Reviewer 2 Report

Dear Authors,

I suggest you keep trying.